DATA RELEASE

# The genome of the mustard hill coral, *Porites astreoides*

Kevin H. Wong[1],* and Hollie M. Putnam[1],*

1 University of Rhode Island, Department of Biological Sciences, USA

## ABSTRACT

Anthropogenic effects have contributed to substantial declines in coral reefs worldwide. However, some corals are more resilient to environmental changes and have increased in relative abundance, thus these species may shape future reef communities. Here, we provide the first draft reference genome for the mustard hill coral, *Porites astreoides*, collected in Bermuda. DNA was sequenced via Pacific Biosciences (PacBio) HiFi long-read technology. PacBio read assembly with FALCON UnZip resulted in a 678-Mbp assembly with 3051 contigs with an N50 of 412,256 and the BUSCO completeness analysis resulted in 90.9% of the metazoan gene set. An *ab initio* transcriptome was also produced with 64,636 gene models with a transcriptome BUSCO completeness analysis of 77.5% versus the metazoan gene set. Functional annotation was completed for 86.6% of proteins. These data are valuable resources for improving biological knowledge of *P. astreoides*, facilitating comparative genomics for corals, and supporting evidence-based restoration and human-assisted evolution of corals.

**Subjects** Genetics and Genomics, Animal Genetics, Marine Biology

## DATA DESCRIPTION

Coral reef ecosystems provide disproportionately large economic, scientific, and cultural value for their global footprint. These ecosystem engineers build coral skeletons that are home to various marine life. This engineering capacity is associated with the nutritional symbiosis between the coral host and their dinoflagellate algal symbionts, Symbiodinaceae [1], which provides excess carbon to fuel coral metabolic processes [2] and stimulate growth of the 3D skeletal structure. Under thermal stress as little as 1 °C above local summer maxima there is a breakdown in the symbiosis of corals and Symbiodinaceae [3], which can result in mass mortality that can reshape reef communities [4].

In the face of rapid climate change, reef-building corals have been under unprecedented stress, resulting in global population declines of these fundamental species [5]. While some species are sensitive to climate change (i.e., ecological 'losers'), others are more resistant or resilient under a rapidly changing environment (i.e., ecological 'winners') [6]. These taxa can often grow and reproduce in a 'weedy' fashion, thus increase in relative abundance on the reef [7]. One such weedy coral species is the mustard hill coral, *Porites astreoides* (Lamarak, 1826). This is a ubiquitous shallow Western Atlantic coral present across large environmental clines from mesophotic depths of ~45 m [8] to shallow mangrove environments [9], and a latitudinal range from Brazil [10] to Bermuda [11]. In contrast to other Caribbean corals, *P. astreoides* has increased in abundance in recent years, with high juvenile [12] and adult [13–15] abundances, but smaller colony sizes [16]. *P. astreoides* is

**Submitted:** 25 April 2022

\* Corresponding authors. E-mail: kevin_wong1@uri.edu; hputnam@uri.edu

Preprint submitted at https://doi.org/10.1101/2022.07.01.498470

considered a 'weedy' species' as it is a hermaphroditic, brooding coral species with a prolonged planulation period [17, 18] of pelagic larvae that exhibit high phenotypic plasticity [19–23] and high recruitment rates [8, 21]. Additionally, larvae and juvenile life stages have been well studied under ocean acidification [24, 25] and temperature stress conditions [22, 26–29] owing to the ease of brooded larval collection. Adult colonies also display phenotypic plasticity in response to different reef environments [9, 23, 30–35] and recovery from thermal bleaching events [23, 31, 32, 36–41].

## Context

As 'omics approaches have emerged, the study of *P. astreoides* has increased for genetic connectivity and population structure [42–44], microbiome community [45–52], Symbiodinaceae community [42, 53, 54], gene expression [32, 34, 50, 55], and epigenetics [56]. Although three *de novo* transcriptome assemblies are available for this species [32, 34, 55] they have relatively low coverage of the anticipated gene repertoire (e.g., BUSCO scores are only 18.1–26.5% complete with respect to the single copy metazoan reference gene set). Therefore, the field is currently limited by the lack of an available reference genome and improved transcriptome. These resources would greatly enhance studies that are reliant on a reference genome; for example, whole genome bisulfite sequencing and genome-wide association studies. Our study is the first to generate a publicly available, assembled, and structurally and functionally annotated reference genome of *P. astreoides*, in addition to an improved reference transcriptome.

## METHODS

## Coral collection, treatment, and sampling

One adult *P. astreoides* colony (Figure 1A) was collected on June 12, 2017, from Bailey's Bay Reef Flats (32° 22′ 27″N, 64° 44′ 37″W) in Bermuda and transported to the Bermuda Institute of Ocean Sciences. The colony was fragmented into genetic replicates using a drill and 3.5-cm diameter size hole saw to generate circular cores of tissue and skeleton (Figure 1B). Replicate fragments were affixed to plugs using underwater epoxy (HoldFast Epoxy Stick, Instant Ocean), which covered the exposed skeletal surface. The replicate fragments were held in indoor tanks with flowing seawater with LED lights (Arctic-T247 Aquarium LED, Ocean Revive) under ambient conditions for 14 days (28 °C with a 12:12 h light cycle at ~115 μmol photons) and then exposed to either ambient or heated conditions (31 °C with a 12:12 h light cycle at ~115 μmol photons). These conditions were applied for 59 days to reduce the concentration of endosymbiotic dinoflagellates (Symbiodiniaceae) and thereby to enrich for host DNA for downstream DNA extraction and sequencing. Coral fragments were immediately snap-frozen in liquid nitrogen and stored at −80 °C on August 28, 2017. An additional four fragments were sampled for RNA sequencing. Of these fragments, two were fragments under ambient conditions; one fragment that experienced the 59-day thermal stress with an additional hyposalinity stress (approximately 18 psu) 30 minutes before snap-freezing, and one fragment under ambient conditions from a different *P. astreoides* colony from the same reef site.

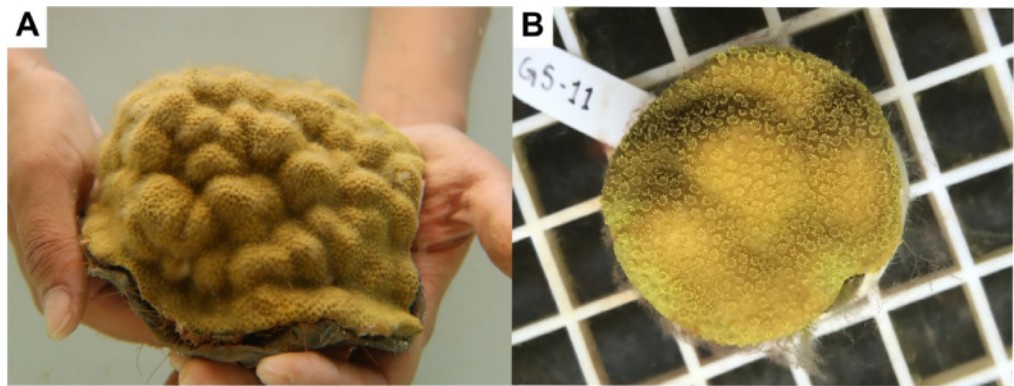

**Figure 1.** (A) Colony of *P. astreoides* and (B) replicate fragment cores used for exposures and extractions.

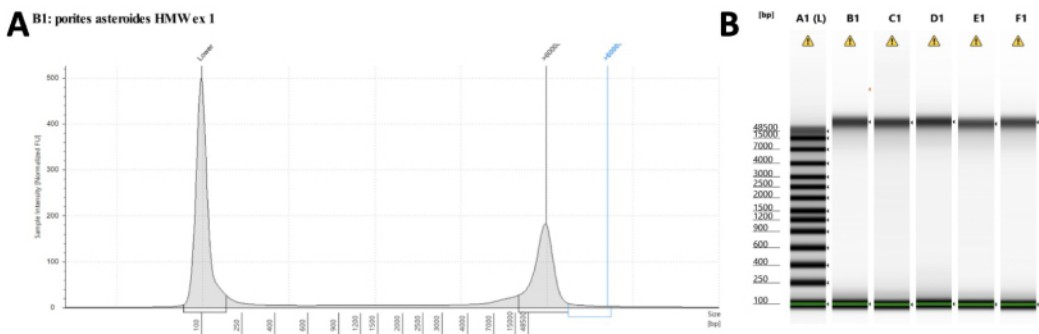

**Figure 2.** Quality check of high-molecular-weight DNA extracted and used for SMRTbell library preparation indicates (A) large-size fragments as shown by the TapeStation peak view and (B) high molecular weight across all extractions.

## Extraction of genomic DNA

The frozen coral samples were homogenized with a mortar and pestle and liquid nitrogen. Coral high-molecular-weight genomic DNA (gDNA) was extracted using the QIAGEN Genomic-tip 100/G (Cat #10223), the QIAGEN Genomic DNA Buffer Set (Cat #19060), QIAGEN RNase A (100 mg/mL concentration; Cat #19101), QIAGEN Proteinase K (Cat #19131), and DNA lo-bind tubes (Eppendorf Cat #022431021) according to the manufacturers' instructions for preparation of tissue samples in the QIAGEN Genomic DNA Handbook (version 06/2015) using five individual sample preps of the same homogenized material to maximize yield. gDNA concentration was quantified using the Qubit DNA Broad Range kit (Invitrogen, Cat #Q32850) and DNA integrity was measured using a Genomic D5000 Screentape on an Agilent TapeStation 4200 system (Figure 2).

## DNA sequencing library, read filtering, and genome assembly

gDNA was sent to Genewiz (Azenta Life Sciences) for library preparation and sequencing using PacBio Sequel I long read technology. A single SMRTbell sequencing library (double-stranded DNA template with hairpins at each end) was constructed. Briefly, the DNA was fragmented by shearing, and DNA damage within the strand and at the ends of the fragments was repaired and the sample purified with AMPure PB beads. Hairpin adapters



**Table 1.** Contig length and assembly statistics.

| Statistics | Filtered assembly | Unfiltered assembly |
|---|---|---|
| No. contigs (≥0 bp) | 3051 | 3095 |
| No. contigs (≥1000 bp) | 3051 | 3095 |
| No. contigs (≥5000 bp) | 3051 | 3095 |
| No. contigs (≥10,000 bp) | 3048 | 3093 |
| No. contigs (≥25,000 bp) | 3030 | 3070 |
| No. contigs (≥50,000 bp) | 2866 | 2905 |
| Largest contig (bp) | 3,369,715 | 2432 |
| Total length (bp) | 677,753,397 | 680,005,989 |
| Total length (≥0 bp) | 677,753,397 | 680,005,989 |
| Total length (≥1000 bp) | 677,753,397 | 680,005,989 |
| Total length (≥5000 bp) | 677,742,019 | 679,999,105 |
| Total length (≥10,000 bp) | 677,611,785 | 679,826,819 |
| Total length (≥25,000 bp) | 674,350,533 | 676,554,116 |
| Total length (≥50,000 bp) | 658,123,657 | 658,911,805 |
| N50 | 412,256 | 408,687 |
| N75 | 189,769 | 190,323 |
| L50 | 437 | 444 |
| L75 | 1049 | 1055 |
| GC (%) | 39.12 | 39.14 |

Bp: base pair.

**Table 2.** Comparison of genome assembly and annotation metrics of *Porites* congeners.

| Feature | *Porites rus* | *Porites lutea* | *Porites australiensis* | *Porites astreoides* |
|---|---|---|---|---|
| Study | [61] | [57] | [62] | This study |
| Assembly size (Mbp) | 470 | 552 | 576 | 678 |
| No. protein coding genes | 39,453 | 31,126 | 30,301 | 64,646 |
| Mean no. of exons per gene | 5.3 | 6.6 | 7.6 | 4.9 |
| Mean exon length (bp) | 239 | 275 | 346 | 190 |
| Mean intron length (bp) | 1224 | 1146 | 1190 | 863 |
| Metazoan BUSCOs (%(gene number), total = 954 genes) | | | | |
| Complete single-copy | 65.5 (625) | 91.9 (877) | 86.6 (826) | 77.5 (739) |
| Complete duplicated | 2.3 (22) | 1.8 (17) | 3.1 (30) | 13.4 (128) |
| Fragmented | 18.7 (178) | 2.8 (27) | 5.8 (55) | 4.1 (39) |
| Missing | 13.5 (129) | 3.5 (33) | 4.5 (43) | 5.0 (48) |

Bp: base pair; Mbp: megabase pair.

were then ligated to the DNA fragment ends and the sample again purified with AMPure PB beads. Sequencing primers were then annealed to form polymerase-primer complexes and the resulting library was sequenced using four SMRT cells on the PacBio Sequel I at GeneWiz (Table 1).

Prior to assembly, reads were mapped to the *Porites lutea* genome [57] with MiniMap v2 (RRID:SCR_018550) [58] and SAMtools v1.9 (RRID:SCR_002105) [59] to exclude non-coral reads. The filtered reads were assembled with Falcon v2.2.4 (RRID:SCR_016089) [60] at Genewiz. To assess genome assembly completeness, the *P. astreoides*, *Porites rus* [61], *P. lutea* [57], and *Porites australiensis* [62] genomes were searched against 954 universal metazoan single-copy orthologs in the metazoa_odb10 gene set using BUSCO v5.2.2 in 'genome' mode (RRID:SCR_015008) [63] (Table 2).

**Table 3.** Comparison of BUSCO metrics of transcriptome completeness across *Porites astreoides* transcriptome assemblies relative to the BUSCO metazoa_odb10 gene set. Transcript quantification was conducted on isogroups using custom scripts for studies with a single asterisk (*) and on transcripts via Trinity for studies indicated with double asterisks (**).

| Metazoan BUSCOs % (gene number), total = 954 genes | Kenkel *et al.* [31] | Mansour *et al.* [55] | Walker *et al.* [34] | Wong and Putnam (this study) |
|---|---|---|---|---|
| Transcripts | 31,663* | 129,718** | 918,990** | 387,749** |
| Complete single-copy | 22.2 (219) | 18.1 (173) | 26.5 (253) | 13.9 (133) |
| Complete duplicated | 0.7 (7) | 12.2 (116) | 36.6 (349) | 7.5 (72) |
| Fragmented | 31.1 (297) | 4.9 (47) | 21.6 (206) | 36.7 (350) |
| Missing | 46.0 (438) | 64.8 (618) | 15.3 (146) | 41.9 (399) |

## Structural and functional annotation

Structural annotation of the *P. astreoides* genome was completed on the University of Rhode Island High Performance Computer 'Andromeda'. As input for MAKER v3.01.03 (RRID:SCR_005309) [64] we used an existing *P. astreoides* transcriptome from samples collected in the Florida Keys, USA [32] and existing congener *P. lutea* peptide sequences from a sample collected in Australia [57]. These files were used as input for an initial round of MAKER to predict gene models directly from this transcriptomic and protein data, respectively. The first round of MAKER included RepeatMasker (RRID:SCR_012954) [65]. The output from the initial MAKER round was used to train *ab initio* gene predictors SNAP (RRID:SCR_002127) [66] and AUGUSTUS (RRID:SCR_008417) [67] using BUSCO v5.2.2(RRID:SCR_015008) [63]. A second round of MAKER was performed using the general feature format (GFF) output of the first round containing the information on the locations of repetitive elements for masking, as well as the locations of expressed sequence tags (ESTs) and proteins and *ab initio* gene prediction from the SNAP and AUGUSTUS outputs. Another round of *ab initio* gene prediction was performed on the output from the second round of MAKER. A third and final round of MAKER was conducted, including training information from the GFF files generated from the second round of MAKER and *ab initio* gene prediction as input. Genome structural annotations were compared against all four *Porites* congeners using AGAT v0.8.1 (Table 2).

The *ab initio* transcriptome generated from the final round of MAKER was compared with other *de novo P. astreiodes* transcriptome assemblies [32, 34, 55] by assessing statistics from BUSCO v5.2.2 in 'transcriptome' mode referencing the metazoa_odb10 gene set [63] (Table 3). The protein set produced from the final round of MAKER was annotated by identifying homologous sequences using BLASTp (Basic Local Alignment Search tool, RRID:SCR_001010, e-value cut-off =1 × 10⁻⁵) against a hierarchical approach of the following databases: (1) UniProt-SwissProt [68], (2) UniProt-TrEMBL [68] and (3) the National Center for Biotechnology Information non-redundant (nr) database [69]. From the BLASTp searches, BLAST2GO (RRID:SCR_005828) [70] was used to characterize putative gene functionality in addition to InterProScan (RRID:SCR_005829) [71].

## Extraction of Total RNA

Total RNA was extracted using the Duet DNA/RNA Miniprep Plus Kit (Zymo Research, Irvine, CA, USA, Cat #D7003) with modifications to the manufacturer's sample preparation steps, as described here. Sterile clippers were used to remove an approximately 1-cm diameter region from the coral tissue. The tissue clipping was placed in 500 µL of DNA/RNA shield (Zymo Cat #R1100-50) with 0.5 mm glass beads (Fisher Scientific Cat #15-340-152) and

homogenized by vortexing for 1 minute. The supernatant (450 µl) was transferred to a new tube and centrifuged at 9000 rcf for 3 min to remove any debris. The supernatant (300 µl) was mixed with 30 µl of Proteinase K (Zymo Cat #D3001-2-20) digestion buffer and 15 µl of Proteinase K in a new tube using a brief vortex step. Equal volume of Zymo lysis buffer (Zymo Cat #D7001-1-200) was added (345 µl) and extraction was completed as outlined in the manufacturer's protocol. RNA concentration was quantified using the Qubit RNA Broad Range kit (Invitrogen Cat #Q10211) and integrity was assayed with an Agilent TapeStation 4200 system and the RNA Integrity Number (RIN) values ranged from 7.0–8.9. RNA libraries were prepared using the Zymo-Seq RiboFree Total RNA Library Kit (Zymo Cat #R3000) according to the manufacturer's instructions, starting with 125 ng of RNA as input. Prepared libraries were sent to Zymo Research Corporation for sequencing using the Illumina HiSeq for paired end sequencing (2 × 100 base pairs [bp]).

### RNA-seq de novo assembly and genome mapping

Four paired-end Illumina libraries generated 35,505,857 raw read pairs. The raw RNA-seq data was quality checked by using fastQC v0.11.8 (RRID:SCR_014583) [72] and visualized with MultiQC v1.7 (RRID:SCR_014982) [73]. Illumina adapters were removed and sequences with a phred score below 30 were trimmed using fastp v0.19.7 (RRID:SCR_016962)[74]. After trimming, 85–93% of the data were kept across the four libraries and the trimmed reads were used for *de novo* assembly and was completed using Trinity v2.9.1 (RRID:SCR_013048)[75] using default parameters except the fastq sequence type and CPU 20 parameters. The trimmed RNA-seq reads were also aligned to the *P. astreoides* genome using STAR v2.7.2b ( RRID:SCR_004463) [76].

### Genome structural and functional annotation

The initial round of MAKER predicted 58,308 putative gene models, the second round 68,481 gene models, and 64,636 were predicted gene models in the third and final round. From our final structural annotation, *P. astreoides* had an average of 4.9 exons per gene, a mean exon length of 190 bp, and a mean intron length of 863. This is comparable to other *Porites* congeners, as shown in Table 2. Of the 64,636 protein encoding genes, 86.6% (*n* = 55,957) were annotated with 47.1% (*n* = 30,444) receiving hits from the SwissProt database, 37.7% (*n* = 24,359) from the TrEMBL database, and 1.8% (*n* = 1154) from the NCBI NR database. A total of 13.4% (*n* = 8679) had no hits to any of the databases. Through BLAST2GO (RRID:SCR_005828) [70], 30,284 genes were assigned putative protein functions against the SwissProt database, and 24,089 were assigned with InterProScan (RRID:SCR_005829) [71]. After removal of duplicate gene functions, 67% (*n* = 43,154) had assigned protein functions using BLAST2GO [70] and InterProScan [71].

### Genome assembly statistics

The sequencing yielded just under 1,000,000 polymerase reads and a total polymerase read length of 20,188,576,450 (Table 1) from a single gDNA library. After quality control and assembly, we obtained a reference genome with a total size of ~678 megabase pairs (Mbp; Table 2). This *P. astreoides* genome had comparable assembly statistics to three other *Porites* species, *P. rus* [61] (assembly size = 470 Mbp), *P. lutea* [57] (assembly size = 552 Mbp) and *P. australiensis* [62] (assembly size = 576 Mbp) (Table 2).



## DATA VALIDATION AND QUALITY CONTROL

Genome assembly completeness was assessed by searching 954 universal metazoan single-copy orthologs in the metazoa_odb10 gene set using BUSCO v5.2.2 [63]. For the *P. astreoides* genome, 739 (77.5%) complete single-copy orthologs were identified, 128 (13.4%) orthologs were completed but duplicated, 39 (4.1%) orthologs were identified but fragmented, and 48 (5.0%) were missing. Compared to other *Porites* species, our *P. astreoides* assembly identified more complete single-copy orthologs than *P. rus* [61] (625 [65.5%] single-copy orthologs) but less than *P. lutea* [57] (877 [91.9%] single-copy orthologs) and *P. australiensis* [62] (826 [86.6%] single-copy orthologs) (Table 2). BUSCO analysis includes 13.4% duplication, which suggests the presence of haplotigs that were not fully removed prior to or during the assembly process and should be addressed in future versions of the assembly.

To describe the mapping potential of the draft *P. astreoides* genome, four paired-end RNA-seq libraries were mapped to the *ab initio* reference genome using STAR v2.7.2b [76]. From the four RNA-seq libraries, mapping percentages ranged from 87.9% to 79.07%, with 17.9–17.05% of the reads mapping to multiple loci. This suggests that we have a suitable *ab initio* reference genome for RNA-seq data for Bermudian populations of *P. astreoides*.

Comparing our *de novo* assembly statistics to those previously published (Table 3) indicates that all *de novo* assemblies to date suffer from transcript fragmentation or duplication versus the BUSCO metazoan reference set. For example, even in the *de novo* transcriptome by Walker *et al.* [34] with the highest complete and single copy BUSCO score and the lowest number of missing reference genes, the values are still only 26.5% complete and single copy and 15.3% remain missing.

While insufficient gene model prediction is possible in our draft assembly, the high number of gene models is probably a result of duplicated contigs from different haplotypes and a fragmented assembly, which is also seen currently in the *de novo* transcriptome. This issue of high predicted gene model number in *P. astreoides* should be improved in subsequent draft assemblies by purging haplotigs and using combined short-read sequence data with our PacBio scaffolds. Here, we have generated a valuable resource for the community, which can now be improved through an iterative process through the coral research community investment.

## REUSE POTENTIAL

The ecological increase in relative abundance of *P. astreoides* in the Atlantic [13] means it is crucial to understand the mechanisms leading to resilience under predicted climate change conditions. Given the potential to improve current transcriptomes [32, 34, 55] and the lack of genomic resources for this species, we provide the first draft reference genome for *P. astreoides*. Although we acknowledge this genome can and should be improved, the goal was to provide a community resource for the advancement of molecular biology and comparative genomics of reef-building corals. *P. astreoides* and the whole *Porites* genus are challenging coral species for molecular biology approaches. *Porites* species have high mucus [77] and lipid [78, 79] content, thick tissues [80, 81], and high endosymbiont densities [82]. This can complicate DNA and RNA extractions by retaining molecules that act as inhibitors in PCR; thus, often, poor quality libraries and *de novo* transcriptome assemblies are generated. The *ab initio* reference genome, transcriptome, and updated annotations generated by this study therefore may serve as a useful resource to the coral field and wider marine invertebrate community.



## DATA AVAILABILITY

Sequencing data is available in NCBI via Bioproject number PRJNA834048, and Biosample numbers SAMN28031657 and SAMN28031658.

The genome assembly and all raw sequencing reads, including the PacBio HiFi reads and Illumina RNA-seq, annotations, alignments, and other results, are available via the Open Science Framework Repository [83].

## DECLARATIONS
## LIST OF ABBREVIATIONS

BLAST: Basic Local Alignment Search Tool; bp: base pair; BUSCO: Benchmarking Universal Single-Copy Orthologs; gDNA: genomic DNA; Mbp: megabase pair; NCBI = National Center for Biotechnology Information; PCR: polymerase chain reaction.

## ETHICAL APPROVAL

Corals were collected under permit 17060807 from the Government of Bermuda Department of Environment and Natural Resources and exported with CITES permit number 19BM0011.

## CONSENT FOR PUBLICATION

Not applicable.

## COMPETING INTERESTS

The authors declare that they have no competing interests.

## FUNDING

This study was funded by a University of Rhode Island's Committee for Research and Creative Activities Research Proposal Development Grant 2017-2018 (awarded to HMP); the Heising-Simons Foundation (awarded to HMP), the Natural Sciences and Engineering Research Council of Canada Post Graduate Scholarship Doctoral (NSERC PGSD3 Award #545967-2020; awarded to KHW), and the Canadian Associates of the Bermuda Institute of Ocean Science (awarded to KHW).

## AUTHORS' CONTRIBUTIONS

Conceptualization: HMP, Methodology HMP, KHW; Investigation: KHW, HMP; Formal analysis: KHW; Resources: HMP; Data Curation: KHW, HMP; Writing Original Draft: KHW, HMP; Writing Review & Editing: KHW, HMP; Visualization: KHW; Supervision: HMP; Funding acquisition: HMP.

## ACKNOWLEDGEMENTS

We would like to acknowledge the Bermuda Institute of Ocean Sciences facilities and the University of Rhode Island High Performance Computing for their logistical support of this work. Additionally, we would like to thank Dr. Samantha de Putron, Dr. Gretchen Goodbody-Gringley and Alexander Chequer for field assistance, Margaret Schedl for laboratory assistance, and Kevin Bryan at the University of Rhode Island High Performance Computing for his assistance with computational needs.



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
