## [Reviewer Report]

Comments on revised manuscriptI appreciate the authors’ effort to address all referee comments. I believe the data will be valuable for the research community.

---

## [Reviewer Report]

Reviewer name and names of any other individual's who aided in reviewer Takeshi TakeuchiDo you understand and agree to our policy of having open and named reviews, and having your review included with the published papers. (If no, please inform the editor that you cannot review this manuscript.)YesIs the language of sufficient quality?YesPlease add additional comments on language quality to clarify if needed
Are all data available and do they match the descriptions in the paper? YesAdditional CommentsAre the data and metadata consistent with relevant minimum information or reporting standards? See GigaDB checklists for examples <a href="http://gigadb.org/site/guide" target="_blank">http://gigadb.org/site/guide</a>YesAdditional CommentsIs the data acquisition clear, complete and methodologically sound?YesAdditional CommentsIs there sufficient detail in the methods and data-processing steps to allow reproduction?NoAdditional Comments1) How many high-quality reads/nucleotides were retained after filtering and applied to the Falcon assembler? The authors also need to describe the parameters for the Falcon.
2) How did the authors manage the duplicated contigs from different haplotypes in the assembly?
3) In Table 2, stas for "scaffolds" are shown. But there is no description of the scaffolding process.Is there sufficient data validation and statistical analyses of data quality? NoAdditional Comments4) In Table 3, the authors should not compare transcriptome (refs 32, 55, and 34) and gene models (this study). Did the authors produce transcriptome assembly from the RNA-seq data in fact? If so, please describe the method for the transcriptome assembly.
5) Results of BLAST2GO and InterProScan were not described.Is the validation suitable for this type of data?NoAdditional CommentsIs there sufficient information for others to reuse this dataset or integrate it with other data?NoAdditional CommentsAny Additional Overall Comments to the AuthorThe number of gene models (64,636) is much higher than those of other Porites species (30,000-40,000). The number of exons per gene is considerably lower than others. These results indicate that the gene models are fragmented, possibly due to insufficient gene model prediction. This issue needs to be discussed.

In the Abstract, the genome size "667 Gbp" should be "667 Mbp." In Table 2 and the main text, the assembly size is 678Mbp. Which is correct?RecommendationMinor Revision

---

## [Reviewer Report]

Reviewer name and names of any other individual's who aided in reviewer Jong BhakDo you understand and agree to our policy of having open and named reviews, and having your review included with the published papers. (If no, please inform the editor that you cannot review this manuscript.)YesIs the language of sufficient quality?YesPlease add additional comments on language quality to clarify if needed
Are all data available and do they match the descriptions in the paper? YesAdditional CommentsAre the data and metadata consistent with relevant minimum information or reporting standards? See GigaDB checklists for examples <a href="http://gigadb.org/site/guide" target="_blank">http://gigadb.org/site/guide</a>YesAdditional CommentsIs the data acquisition clear, complete and methodologically sound?YesAdditional CommentsIs there sufficient detail in the methods and data-processing steps to allow reproduction?YesAdditional CommentsIs there sufficient data validation and statistical analyses of data quality? YesAdditional CommentsIs the validation suitable for this type of data?YesAdditional CommentsIs there sufficient information for others to reuse this dataset or integrate it with other data?YesAdditional CommentsAny Additional Overall Comments to the AuthorPorites astreoides is an important coral species and this reviewer thinks. 
All the major reference construction parameters have shown a high quality assembly.

Predicted gene number, 64,636, is a bit too high. This needs to be checked and improved. (This number has been fluctuating. Not critical, though)



RecommendationAccept